# Bladder Management Strategies for Urological Complications in Patients with Chronic Spinal Cord Injury

**DOI:** 10.3390/jcm11226850

**Published:** 2022-11-20

**Authors:** Yu-Chen Chen, Yin-Chien Ou, Ju-Chuan Hu, Min-Hsin Yang, Wei-Yu Lin, Shi-Wei Huang, Wei-Yu Lin, Chih-Chieh Lin, Victor C. Lin, Yao-Chi Chuang, Hann-Chorng Kuo

**Affiliations:** 1Graduate Institute of Clinical Medicine, College of Medicine, Kaohsiung Medical University, Kaohsiung 80708, Taiwan; 2Department of Urology, Kaohsiung Medical University Hospital, Kaohsiung Medical University, Kaohsiung 80708, Taiwan; 3Department of Urology, National Cheng Kung University Hospital, College of Medicine, National Cheng Kung University, Tainan 70403, Taiwan; 4Division of Urology, Department of Surgery, Taichung Veterans General Hospital, Taichung 407, Taiwan; 5Department of Urology, Chung Shan Medical University Hospital, Taichung 40201, Taiwan; 6Department of Urology, Taipei Hospital, Ministry of Health and Welfare, New Taipei 242033, Taiwan; 7Department of Urology, National Taiwan University Hospital Yun-Lin Branch, Douliou 640203, Taiwan; 8Department of Urology, Chiayi Chang Gung Memorial Hospital, Chiayi 261363, Taiwan; 9Department of Urology, Taipei Veterans General Hospital, Taipei 112304, Taiwan; 10Department of Urology, School of Medicine, College of Medicine, Shu-Tien Urological Research Center, National Yang Ming Chiao Tung University, Taipei 112304, Taiwan; 11Department of Urology, E-Da Hospital, Kaohsiung 824, Taiwan; 12Department of Urology, Kaohsiung Chang Gung Memorial Hospital, Chang Gung University, College of Medicine, Kaohsiung 833401, Taiwan; 13Department of Urology, Hualien Tzu Chi Hospital, Buddhist Tzu Chi Medical Foundation and Tzu Chi University, Hualien 97004, Taiwan

**Keywords:** spinal cord injuries, urinary catheterization, bladder, neurogenic bladder, complications, self-catheterization

## Abstract

Neurogenic lower urinary tract dysfunction, common in patients with chronic spinal cord injury, inevitably results in urological complications. To address neurogenic lower urinary tract dysfunction after spinal cord injury, proper and adequate bladder management is important in spinal cord injury rehabilitation, with the goal and priorities of the protection of upper urinary tract function, maintaining continence, preserving lower urinary tract function, improvement of SCI patients’ quality of life, achieving compatibility with patients’ lifestyles, and decreasing urological complications. This concise review aims to help urologists address neurogenic lower urinary tract dysfunction by focusing on the risks of long-term urological complications and the effects of different bladder management strategies on these complications based on scientifically supported knowledge.

## 1. Introduction

The annual rate of spinal cord injury (SCI) was estimated by the National SCI Database (2016) to be 54 cases per million people in the United States and about 17,000 new SCI cases per year [1]. There are approximately 282,000 people in the United States affected by SCI [1]. Altered lower urinary tract (LUT) function, known as neurogenic LUT dysfunction (NLUTD) due to central or peripheral neurogenic lesions, frequently occurs secondary to chronic SCI and has an impact on the quality of life (QoL) [2]. The main problems associated with NLUTD in chronic SCI patients are a failure to store in or empty the bladder or a combination of the two [3]. If the bladder and bladder outlet dysfunction are not properly managed, they may consequently lead to urinary tract infection (UTI), urosepsis, poor bladder compliance, upper urinary tract deterioration, renal failure, urinary tract calculi, skin complications, depression (which also complicates urologic treatment), and significant morbidity and mortality in some SCI patients [2]. To address NLUTD after SCI, proper and adequate bladder management is important for SCI rehabilitation, with the goal of preserving the function of the upper urinary tract, maintaining urinary continence, avoiding urological complications, and achieving compatibility with the patient’s lifestyle [2]. Clean intermittent catheterization (CIC) is generally recommended as the optimal method for bladder drainage with the advantages of lower rates of UTI, stones, urinary tract fistula, strictures, and cancers compared to an indwelling catheter [4,5]. Although CIC is reported to be the most commonly (42.6%) used method, some patients still choose other bladder-emptying methods (condom drainage, 11.3%; suprapubic cystostomy, 11.3%; bladder reflex triggering, 11%) because CIC may not be feasible to maintain or perform in all SCI patients based on their comfort, convenience, or maximal independence [6]. Alternative conservative options include indwelling catheters (transurethral catheters or suprapubic cystostomy), condom catheters, as well as bladder reflex triggering or bladder expression with the Valsalva or Credé maneuver. However, assisted bladder emptying with triggered reflex voiding or bladder expression is only suggested when the intravesical remains within 40 cm H_2_O [7]. Other surgical bladder-emptying methods may be considered if necessary. However, despite the improved bladder-emptying treatment options, urologic complications due to NLUTD are still a major cause of morbidity and mortality in SCI patients and remain one of the most important health issues [8,9,10,11]. Although the major goals of treatment and bladder management for NLUTD have been discussed, few papers aim to evaluate the association between different bladder management strategies and the risks of urological complications in chronic SCI patients [12,13,14,15]. Therefore, the purpose of this manuscript was to review the available data to address issues including long-term urological complications in chronic SCI patients, the effects of different bladder management strategies on UTIs, urge urinary incontinence (UUI), poor bladder compliance, the preservation of renal function, the prevention of urolithiasis, and satisfaction with enterocystoplasty in these patients.

## 2. Urological Complications in Chronic SCI Patients

As summarized in Figure 1, NLUTD gradually and inevitably results in urological complications [16,17,18], which are closely related to each other. The rate of urological complications remains high in patients with chronic SCI [12,19]. Furthermore, the level of SCI and urological complications are closely associated [20,21]. As summarized in Figure 2, Chen et al. analyzed urological complications based on different SCI levels and reported that severe UI occurred significantly in patients with cervical and thoracic SCI, whereas urolithiasis was found to be more significant in patients with sacral SCI than in other levels of SCI [12]. Weld et al. analyzed bladder dysfunction based on urodynamic findings, and high rates of poor bladder compliance and high detrusor leak points were found in patients with sacral injuries [22]. However, patients with combined suprasacral and sacral injuries may have relatively unpredictable urodynamic findings and different estimated voiding dysfunction. Therefore, it is important to screen high-risk patients with SCI, especially when the detrusor leak-point pressure is higher than 40 cm H_2_O, indicating that the upper urinary tract is endangered [23,24].

A UTI is the most common reason for SCI patients presenting to the emergency department and being re-hospitalized [25]. UTIs were reported to occur in 100% of patients with SCI in one study with at least a 40-year follow-up [26]. The incidence of UTI was reported to peak in the 1st and 10th five-year intervals [26]. Pickelsimer et al. conducted a 10-year follow-up study, in which UTI, hydronephrosis, and urolithiasis were the three main complications of NLUTD [27]. However, Chen et al. compared the urological complications at different time periods after SCI and found that there was no significant difference in the occurrence rate of urological complications among different SCI durations [12]. Another retrospective study found that the percentage of patients with urolithiasis was 20% and 80% before and after 20 years after SCI, respectively [22]. Overall, most complications initially occurred during the first 25 years after SCI. Close follow-up of UTIs, renal condition, and bladder function is important for all SCI patients and at any disease duration.

## 3. The Effects of Different Types of Bladder Management on UTI Post-SCI

UTIs are common in patients with SCI. UTIs or bacteriuria occur in up to 57% of patients within the 1st year post-SCI [28]. The risk factors for UTIs in SCI patients were reported to be cervical SCI, male sex, and condom or catheterization [26]. Recurrent UTIs may be a sign of urinary tract calculi, pyelonephritis, or NLUTD causing urine stasis (increased residual urine). Combined with poor urodynamic bladder function, UTIs can lead to poor QoL and become approximately 9.5% of the cause of death in SCI patients [29].

Catheter-related UTIs are the most common healthcare-associated infections in the world, and bacteriuria is universal 30 days after catheter placement [30] because catheters facilitate bacteria bypassing host defense mechanisms and entering the bladder directly with ease. However, more than 60% of SCI patients still need some form of bladder management for urination [31]. Therefore, finding the optimal bladder management strategy to maintain bladder function and avoid UTIs is necessary. According to the current guidelines of the European Association of Urology and the American Urological Association, CIC and indwelling transurethral catheters are the most- and least-preferred bladder management methods, respectively, with regard to UTI risk [32,33]. Ned et al. conducted a systematic review, which reported that CIC had a lower UTI rate than indwelling transurethral catheters, but the evidence was inconclusive when comparing transurethral catheters and suprapubic cystostomy [13]. CIC has the advantage of cyclical bladder emptying, which replicates normal bladder function and reduces the number of bacteria by avoiding persistent foreign body placement. However, more patients with an SCI duration of longer than 5 years would choose an indwelling catheter or cystostomy over CIC [12]. Improvements in patient education, regular surveillance of the urinary tract, and correct medical treatment are necessary to reduce the rate of urological complications and improve the QoL of chronic SCI patients.

In a large cohort of SCI patients in Taiwan, a significantly higher UTI rate was observed in patients undergoing bladder management other than normal voiding [12]. Overall, we suggest that SCI patients should be treated with CIC rather than a chronic indwelling catheter if they cannot empty fully [14]. If possible, SCI patients with a leak-point pressure lower than 40 cm H_2_O should be trained to void spontaneously by abdominal straining or percussion to lower the rate of UTI or urological complications.

It is imperative to treat febrile UTIs with adequate antibiotics based on urine culture results. Asymptomatic bacteriuria should not be treated according to the consensus reached by the National Institute on Disability and Rehabilitation Research Group [34]. Urodynamic studies should be arranged for SCI patients with recurrent UTIs. When elevated intravesical pressure, large residual urine, vesicoureteral reflux, contracted bladder, or other LUT abnormalities are detected, medication or surgical intervention to reduce intravesical pressure, increase bladder capacity, or an anti-reflux procedure should be considered.

Recurrent UTIs in SCI patients may indicate suboptimal management of NLUTD and its subsequent urological complications. Besides botulinum toxin A (BoNT-A) for NDO [35], the removal of bladder stones, and avoiding indwelling catheters if possible [36], there are still limited preventive strategies. Regarding prophylactic antibiotics, daily antibiotic prophylaxis is not suggested for the prevention of recurrent UTIs [37], whereas weekly oral cyclic antibiotics (WOCA) showed significantly reduced UTIs without the emergence of bacterial resistance compared to the control group [38,39]. Probiotics may be a potential strategy to reduce recurrent UTIs; however, in a recent systemic review [40], only two studies showed that probiotics could reduce the risk of recurrent UTIs and the remainder demonstrated inconclusive results. Antibiotic bladder instillations, such as neomycin-polymyxin or gentamicin, can decrease the frequency of symptomatic UTIs in neurogenic bladder patients on CIC, without increasing multidrug resistance in UTI organisms [41,42]. Intravesical hyaluronic acid instillation is efficient and safe for patients with a neurogenic bladder [43,44]. However, further large, randomized studies were needed to comment on the effect of the WOCA, probiotics, and bladder instillations of either antibiotics or hyaluronic acid. The EAU could not find sufficient evidence to recommend other prevention methods for UTIs such as cranberry juice, methenamine Hippurate, L-methionine, estrogen supplementation, or D-mannose in patients with neuro-urological symptoms [32,45,46,47,48].

## 4. Bladder Management of Urge Urinary Incontinence (UUI) in Chronic SCI Patients

In an assessment of 236 patients with a mean follow-up of 24 years, 43% of patients reported UUI, with paraplegics reporting daily incontinence more frequently than tetraplegics (presumably because of catheter dependence in the latter group) [48]. Only 19% of patients used some form of medication for assistance in managing their incontinence. Surprisingly, CIC was associated with higher rates of UUI than other types of bladder management. In a study by Blanes et al., which included 60 patients with traumatic paraplegia, the complication rate of UUI was found to be more than twice that found in a previous report [49]. Current evidence shows that the effects on bladder function depend on the different levels and locations of SCI [20], which may potentially explain the different rates reported by these two studies.

The appropriate management of NLUTD in patients with SCI is a major challenge for urologists. In most patients with suprasacral SCI who have neurogenic detrusor overactivity (NDO) with or without detrusor sphincter dyssynergia (DSD), bladder management by patients themselves depends on good hand dexterity, powerful abdominal muscle strength, intact bladder sensation, and coordination of the urethral sphincter during stimulation to voiding [22]. Regarding the medication for NDO, antimuscarinics are the most common treatment and are suggested as the first-line treatment by current guidelines [32]. The role of Beta-3-adrenergic receptor agonists, which are not yet approved by the FDA for the treatment of neurogenic bladder, is still unclear [50]. In a recent systemic review, mirabegron was shown to improve the storage symptoms of NLUTD and urologic QoL with very few side effects [51]. Improved maximum cystometric capacity and bladder compliance were demonstrated after treatment with mirabegron in only two studies with short follow-ups [52,53]; however, other studies showed no significant changes in the urodynamic parameters [54,55]. Vibegron, another novel Beta-3-adrenergic receptor agonist, was reported to improve bladder capacity and bladder compliance without apparent adverse effects in SCI patients with NLUTD in two recent studies with limited cases [56,57]. Further prospective studies are necessary for the role of Beta-3-adrenergic receptor agonists in the treatment of NDO.

Currently, it is possible to use a botulinum toxin A (BoNT-A) injection over the bladder detrusor to decrease detrusor contractility [58,59]; a BoNT-A injection over the urethral sphincter to decrease urethral resistance [60,61]; or combine detrusor and urethral BoNT-A injections to spontaneously improve bladder storage and emptying [62]. Although the efficacy of BoNT-A over the sphincter was reported to be high with few side effects, this treatment is not licensed and the necessity of reinjection is its main disadvantage [60,63,64]. In addition, because of limited evidence from small studies, further randomized studies are still needed to comment on the effectiveness of BoNT-A and the optimal dose [65].

## 5. Different Types of Bladder Management of Poor Bladder Compliance

Poor bladder compliance, which is an abnormal relationship between urine volume and intravesical pressure, leads to a gradual increase in intravesical pressure during the bladder-filling phase [15]. When the bladder capacity decreases and bladder pressure increases, the risk of upper urinary tract deterioration increases [15]. Therefore, bladder capacity and compliance are the two main factors in the management of SCI patients because persistently high intravesical pressure can adversely affect the function of the ureter and ureterovesical junction [66]. Poor bladder compliance is the most common cause of hydronephrosis in SCI patients, and grade 3–4 hydronephrosis is also common in patients with poor bladder compliance [67].

Poor bladder compliance was reported in approximately 28–78.6% of SCI patients, which mostly occurred in those with sacral injuries [15,68,69]. After physical rehabilitation for SCI has stabilized, bladder compliance might worsen and the urinary tract might deteriorate with time. Regarding different bladder management strategies, poor bladder compliance was found in 35% of SCI patients with spontaneous voiding, 26% with CIC, and 77% with chronic indwelling catheters [15]. Overall, chronic SCI patients with bladder management involving CIC or spontaneous voiding could sustain normal bladder compliance. In contrast, poor bladder compliance or its development over time was associated with SCI patients using chronic transurethral catheters [15]. Normal bladder compliance was more common in patients with suprasacral than sacral injuries, and with incomplete injuries than complete SCI, independent of the bladder management method [22].

For patients with poor bladder compliance and upper urinary tract deterioration, recurrent febrile UTIs, or severe UUI due to neurogenic detrusor overactivity, first-line treatment includes anticholinergics or antispasmodics [32]. Urodynamic studies and upper urinary tract assessments should be performed three months later [2]. Patients with normal hand function were advised to use CIC combined with a Credé maneuver at normal intravesical pressure [14]. If hydronephrosis or recurrent UTI develops, intravesical BoNT-A and surgical intervention are recommended to increase bladder capacity, reduce intravesical pressure, or decrease bladder outlet resistance [58]. Klaphajone et al. studied detrusor BoNT-A injections in patients with high intravesical pressure and compliant bladders. Six weeks after treatment, bladder capacity, compliance, and reflex volume significantly increased, whereas intravesical pressure significantly decreased. The content of the bladder capacity increased significantly [70]. These results were maintained at 16 weeks but returned close to baseline at 36 weeks. Overall, BTX-A injections may provide a valuable alternative to radical surgery.

Surgical operations, which included augmented enterocystoplasty (AE) [71], bladder auto augmentation [72], continent and incontinent urinary diversion, and external sphincterotomy [73,74], remain an option when most conservative treatments fail to manage urological complications. Bladder augmentation was suggested only for quadriparetic and paraplegic patients with hand function that is sufficient for CIC. Continent urinary diversions were the creation of pouches or with a catheterizable channel by Mitrofanoff appendicovesicostomy [75]. Continent diversion was mainly indicated in female patients in whom CIC was difficult to perform, in male patients with severely incompetent urethras, and in quadriplegics whose hand function was only sufficient to perform CIC from an abdominal ileostoma. Incontinent urinary diversion included ileal or colon conduits [76]. External sphincterotomy was performed in quadriplegic and quadriparetic patients with severe autonomic dysreflexia or DSD. Patients undergoing external sphincterotomy were unable to manage their daily activities and CIC, whereas patients treated with bladder augmentation were able to participate in social activities and perform CIC. Transurethral resection of the prostate or laser prostatectomy should be considered in elderly patients with benign prostatic obstructions and severe dysuria. Careful investigation and appropriate medication are suggested for patients with severe UUI [32]. In patients with refractory symptoms, surgical procedures are performed using bladder augmentation for NDO, poor compliance, or periurethral Teflon injections for severe urethral incompetence [58,70,71].

For SCI patients who do not want to change their present bladder management strategies for various reasons, such as condom catheterization and an external collecting appliance for incontinence, suprapubic cystostomy, or indwelling urethral catheterization for urinary retention, we suggest close and periodic follow-ups.

## 6. Renal Function Preservation in Chronic SCI Patients

In the past, renal failure with urinary incontinence was indicated to be the major cause of death after SCI, and the rate of renal deterioration-associated mortality in NLUTD was approximately 50% before the development of regular surveillance of the upper urinary tract and LUT function [77]. Although the mortality rate has gradually reduced because of the improved urological management of NLUTD [11,78], renal failure is still an independent risk factor for mortality in SCI patients because renal deterioration in NLUTD is usually a consequence associated with high intravesical pressure, poor bladder compliance, vesicoureteral reflux, hydronephrosis, UTI, and UUI [79]. Clinical guidelines suggest that optimal bladder management is a critical component of any rehabilitation program for patients with SCI [80,81]. Interventional procedures for treatment include intermittent or indwelling catheterization, external catheter use, augmentation cystoplasty, and urinary diversion [82]. In addition, pharmacotherapy can be used to improve bladder emptying, filling, or storage of urine [2]. Alpha-blockers and cholinergic agents are indicated to improve bladder emptying and anticholinergic agents, and BoNT-A is suggested to improve the storage of urinary symptoms [83].

Patients with DSD, poor bladder compliance, and high intravesical pressure at the end-filling bladder phase are at a high risk of renal failure. The rate of chronic renal disease in patients with paraplegia is higher than that in the general population [84]. Urological management affects bladder compliance, which can change over time [15]. CIC is a superior method for preserving bladder compliance and avoiding upper urinary tract complications associated with poor compliance [15]. Patients with DSD currently using an indwelling catheter, performing CIC, or voiding spontaneously should be monitored annually to avoid renal failure. Oral antimuscarinic agents or intravesical Botox injections may reduce bladder pressure and preserve renal function in the long-term treatment of NLUTD [62,70,85].

## 7. The Risk of Urolithiasis in the Different Types of Bladder Management

Approximately 3–16% of SCI patients will initially develop a kidney stone within 10 years of injury [86,87,88]. The risk of the formation of kidney stones appears to be highest within the first three months of SCI (31 cases per 1000 persons per year) and then reduces after eight years following SCI (8 cases per 1000 persons per year) [88]. The median time between SCI and the development of the first renal stone was reported to be 7.5 years [87]. It has been reported that the rate of renal stones is 7–8 times higher in SCI patients than in the general population, and the risk of nephrolithiasis increases over time after SCI [88]. Risk factors for renal stones include UTIs, bacteriuria (especially urea-splitting bacteria), metabolic changes in urine due to calcium mobilization from bones, and reduced urine output due to fluid restriction to reduce the frequency of CIC and UUI. Lane et al. found that over half of the renal calculi in SCI patients (56%) managed their bladder with CIC, whereas the remaining half were divided between urinary diversion (21%), suprapubic cystostomy (18%), and self-voiding (6%) [87]. Overall, UTI prevention, bacteriuria management, and maintaining adequate fluid levels are important for preventing the development of renal stones in SCI patients.

The incidence of bladder stones in SCI patients is variable, ranging from 3% to 36% [89,90,91,92]. Overall, 98% of stones in patients with SCI were reported to be apatite or struvite in composition. There may be two specific time frames for the formation of urolithiasis in SCI patients including the acute phase, when immobilization leads to hypercalciuria, and the chronic phase, which is associated with chronic catheters due to NLUTD [86]. Catheter encrustation has been found to be associated with bladder calculi, with 85% of patients with catheter encrustation shown to have bladder calculi on cystoscopy [93]. Indeed, the incidence of bladder calculi is affected by the type of bladder management. The rate of bladder stones is 9–20 times higher with indwelling catheters (transurethral catheter or suprapubic cystostomy) than with patients with CIC or continent patients who are free of catheters [90,91]. Among the indwelling catheters, the rates of bladder stones in SCI patients with suprapubic cystostomy and transurethral catheters have been reported to be 4–25% and 4–6.6%, respectively, whereas the stone formation rates decreased to 2% and 1.1% in patients with CIC and reflex micturition, respectively [89,90,91]. Regarding the time between SCI and bladder stone development, the shortest time interval (2.6 years) was found in patients with transurethral catheters, followed by those with suprapubic cystostomy (4.9 years), CIC (9.7 years), and reflex voiding (17.6 years) [89]. Furthermore, the risk of recurrence of bladder stone formation increased if an SCI patient previously had a bladder stone. The reasons for this may be pre-existing risk factors [94,95], especially bladder management, which did not change after the removal of the bladder stone. The bladder stone recurrence rate was reported to be 23%, with the highest recurrence rate (40%) in patients with transurethral catheters, followed by those with suprapubic cystostomy (28%), CIC (22%), and reflex voiding (0%) [89]. Overall, the risk of bladder calculi was the highest in SCI patients with indwelling catheters (transurethral catheters and suprapubic cystostomy); therefore, indwelling catheters should be avoided to reduce stone complications. If unavoidable, suprapubic cystostomy may be preferable to transurethral catheterization.

## 8. Long-Term Complications and Satisfaction of Augmentation Enterocystoplasty (AE) in Chronic SCI Patients

Currently, the first-line treatment for NLUTD in SCI is anticholinergic agents, timely voiding schedules, and CIC in relatively good circumstances, followed by detrusor BoNT-A injections when the effects of conservative treatment are inadequate [32]. Repeat BoNT-A injections every 6–9 months are necessary to maintain the therapeutic effects of NDO, especially in patients with chronic SCI [58,96]. If the outcome is still refractory to repeated BoNT-A injections, intravesical pressure will remain high, which eventually leads to hydronephrosis, renal failure, and UUI, so more aggressive surgical treatment should be considered to obtain life-long therapeutic effects instead of periodic BoNT-A injections [71].

AE should be considered in patients with reduced bladder capacity and poor compliance due to refractory NLUTD [97,98]. This procedure is recommended for reconstructing the bladder and increasing the bladder capacity and, therefore, has been used to treat bladder dysfunction in adults and pediatric patients with myelomeningocele [99]. AE can effectively reduce intravesical pressure during bladder storage and increase bladder capacity in patients with end-stage bladder diseases or refractory detrusor overactivity [100]. Although AE is a procedure with long-term durability and high satisfaction, some major complications still exist [2]. Overall, 86.9% of 76 patients who underwent AE were well or moderately satisfied with the treatment outcomes, and the postoperative UI rate was only 16.5% in a large cohort by Wu et al. [101]. Moreover, 76% of patients required CIC, whereas others could void spontaneously with the Credé maneuver. Among the patients who needed CIC, some finally chose an indwelling transurethral catheter or cystostomy for convenient bladder emptying. In addition, AE is associated with a risk of bladder malignancy. A recent systemic review reported that the estimated incidence of developing a malignant tumor after AE ranged from 0 to 272.3 per 100,000 patients/year [102,103,104,105,106,107,108,109,110] and that 51.6% of the malignancy was adenocarcinoma. Up to 90% of bladder malignancies were diagnosed more than 10 years after AE. Although the exact mechanism of carcinogenesis after AE is still unclear, several factors, such as bacteriuria, chronic inflammation, and urinary hyperosmolality conditions, are reported to be possibly involved [111,112,113,114,115,116]. The follow-up time for regular surveillance after AE is controversial; however, an annual cystoscopy starting 10 years after AE was recommended by most studies [102,105,117,118].

AE is usually performed during the final step of NDO treatment due to the relatively high rates of postoperative complications. In a small prospective study comparing the QoL between SCI patients who underwent AE (*n* = 16) and those who underwent repeat BoNT-A injections (*n* = 14), the continence rate and QoL index were both significantly higher in the AE group (continence rate: 87.5% vs. 42.3%, *p* = 0.0187; QoL index: 1.625 vs. 1.077, *p* = 0.037). The overall outcome was good and no patients post-AE had poor bladder compliance or higher intravesical pressure at the filling phase [119]. Therefore, if treatment with BoNT-A detrusor injections is not effective or if patients cannot tolerate frequent detrusor injections, we suggest AE as a final bladder management strategy.

## 9. Conclusions

The ideal bladder management strategy in patients with chronic SCI has been the subject of much debate among urologists. Generally, CIC is the preferred management for SCI patients who fail to empty their bladders effectively [7] and it is suggested that indwelling transurethral and suprapubic catheters be avoided whenever possible. However, the bladder treatment strategy should be flexible and individualized for every patient. The priorities in the management of NLUTD in SCI patients should be the following in order of importance: (1) the preservation of renal function, (2) a reduction in UTIs, (3) efficient bladder emptying, (4) the avoidance of indwelling catheters, (5) patient agreement with the management strategy, and (6) freedom from medication after proper bladder management. Because the diagnosis and management of NLUTD are complicated, patient-centered guidelines for bladder management are also necessary [120]. Although it is well known that bladder symptoms are usually associated with a reduction in patients’ QoL [121], less than half of SCI patients are estimated to have good knowledge of bladder management. Therefore, it is necessary to increase patient awareness of the urological complications associated with NLUTD. Studies evaluating the outcomes and risks of different bladder management strategies provide a unique opportunity to counsel patients based on scientifically supported knowledge.

## Figures and Tables

**Figure 1 jcm-11-06850-f001:**
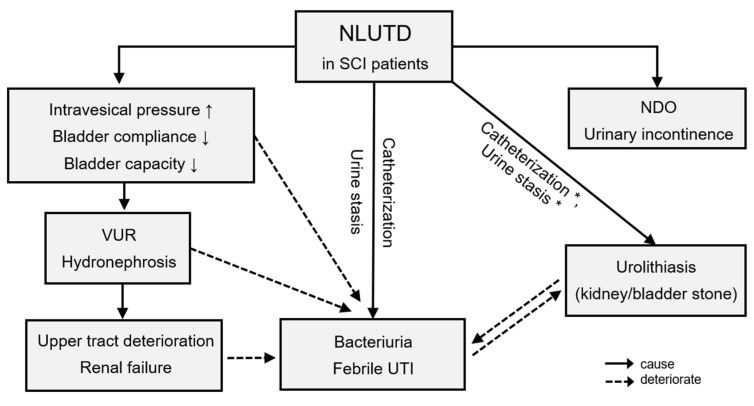
Urological complications in chronic spinal cord injury patients. * Catheterization and urine stasis were mainly related to bladder stones. Abbreviations: SCI, spinal cord injury; NLUTD, neurogenic lower urinary tract dysfunction; VUR, vesicoureteral reflux; NDO, neurogenic detrusor overactivity; UTI, urinary tract infection.

**Figure 2 jcm-11-06850-f002:**
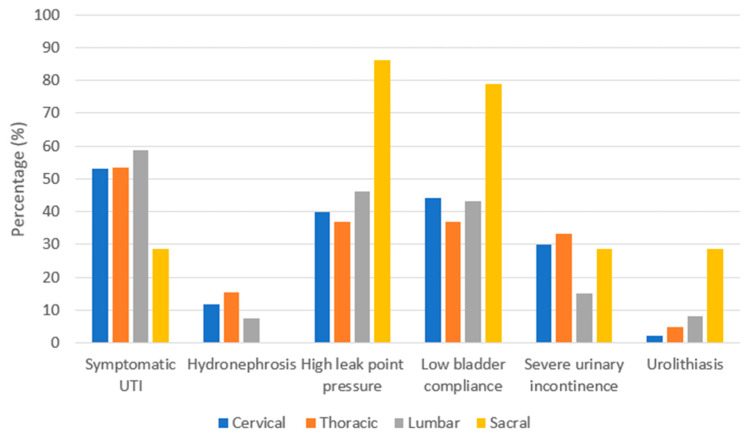
The percentage of urological complications based on different levels of spinal cord injuries [12,22]. Abbreviation: UTI, urinary tract infection.

## Data Availability

Not applicable.

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
