# Peer review of "Bladder Management Strategies for Urological Complications in Patients with Chronic Spinal Cord Injury"

_jcm, 2022, doi:10.3390/jcm11226850_

Round 1
Reviewer 1 Report
Managing bladder dysfunctions in patients with chronic spinal cord injury became an essential part of daily practice. The proper steps that must be included in such patient management are not yet standardized. Proper and adequate bladder management is important in spinal cord injury rehabilitation. The main goal is maintaining continence, decreasing urological complications, preserving upper and lower urinary tract function, and achieving compatibility with the patient’s lifestyle. This material helps urologists to address neurogenic lower urinary tract dysfunction by focusing on the risks of long-term urological complications and the effects of different bladder management strategies based on the knowledge available so far.
It is an important and useful material that could be slightly improved.
Line 127 – add precise numbers for the mortality rate linked to UTIs.
How were the numbers added from the two cited studies to result Figure 2?
Please provide references for lines 55-57.
Please provide references for lines 183-186.
Please provide references for lines 199-210.
Please provide references for lines 230-233.
Please provide references for lines 257-259 and 261-263.
Please provide references for lines 309-316.
Author Response
Dear reviewer:
I am very grateful to your comments for the manuscript. According with your advice, we amended the relevant part in manuscript. The reviewer’s questions were answered below.
Reviewer 1.
Managing bladder dysfunctions in patients with chronic spinal cord injury became an essential part of daily practice. The proper steps that must be included in such patient management are not yet standardized. Proper and adequate bladder management is important in spinal cord injury rehabilitation. The main goal is maintaining continence, decreasing urological complications, preserving upper and lower urinary tract function, and achieving compatibility with the patient’s lifestyle. This material helps urologists to address neurogenic lower urinary tract dysfunction by focusing on the risks of long-term urological complications and the effects of different bladder management strategies based on the knowledge available so far.
It is an important and useful material that could be slightly improved.
Point 1: Line 127 – add precise numbers for the mortality rate linked to UTIs.
Response 1: Thank you for your comment. We have revised the statement to “Combined with poor urodynamic bladder function, UTIs can lead to poor QoL and approximately 9.5% of the cause of death in SCI patients [1].”
Reference 1: Kriz J, Sediva K, Maly M. Causes of death after spinal cord injury in the Czech Republic. Spinal Cord. 2021 Jul;59(7):814-820.
Point 2: How were the numbers added from the two cited studies to result Figure 2?
Response 2: Thank you for your comment. In the first study [1], authors only reported the percentage of clinical findings including symptomatic UTI, hydronephrosis, severe UUI and stones in different levels of spinal cord injury. In the other study [2], the urodynamic findings, including high Pdet and poor bladder compliance among different levels of SCI, were reported.
Reference:
- Sheng-Fu Chen, Yuan-Hong Jiang, Jia-Fong Jhang, Cheng-Ling Lee, Hann-Chorng Kuo. Bladder management and urological complications in patients with chronic spinal cord injuries in Taiwan. Tzu Chi Medical Journal 26 (2014) 25-28.
- Weld KJ, Dmochowski RR. Association of level of injury and bladder behavior in patients with post-traumatic spinal cord injury. Urology. 2000 Apr;55(4):490-4.
Point 3:
Please provide references for lines 55-57.
Response 3: Thank you for your comment. We have provided the references as “Ku JH. The management of neurogenic bladder and quality of life in spinal cord injury. BJU Int 2006; 98: 739-745”.
Point 4:
Please provide references for lines 183-186.
Response 4: Thank you for your comment. We have provided the references as ‘Weld KJ, Graney MJ, Dmochowski RR. Differences in bladder compliance with time and associations of bladder man-agement with compliance in spinal cord injured patients. J Urol. 2000 Apr;163(4):1228-33.’.
Point 5:
Please provide references for lines 199-210.
Response 5: Thank you for your comment. We have provided the references as below in the revised manuscript.
. Weld KJ, Graney MJ, Dmochowski RR. Differences in bladder compliance with time and associations of bladder management with compliance in spinal cord injured patients. J Urol. 2000 Apr;163(4):1228-33.
. Weld KJ, Dmochowski RR. Association of level of injury and bladder behavior in patients with post-traumatic spinal cord injury. Urology. 2000 Apr;55(4):490-4.
. Blok BF, Castro‐Diaz D, Del Popolo G, et al. EAU Guidelines on Neuro‐Urology. European Association of Urology. 2019.
. Schurch B, Stöhrer M, Kramer G, Schmid DM, Gaul G, Hauriet D. Botulinum-A toxin for treating detrusor hyperre-flexia in spinal cord injured patients: a new alternative to anticholinergic drugs? Preliminary results. J Urol 2000;164:692-7.
. Kinnear N, Barnett D, O'Callaghan M, Horsell K, Gani J, Hennessey D. The impact of catheter-based bladder drainage method on urinary tract infection risk in spinal cord injury and neurogenic bladder: A systematic review. Neurourol Urodyn. 2020 Feb;39(2):854-862.
. Ku JH. The management of bladder and quality of life in spinal cord injury. BJU Int 2006; 98: 739-745
Point 6: Please provide references for lines 230-233.
Response 6: Thank you for your comment. We have provided the references as below in the revised manuscript.
. Blok BF, Castro‐Diaz D, Del Popolo G, et al. EAU Guidelines on Neuro‐Urology. European Association of Urology. 2019.
. Schurch B, Stöhrer M, Kramer G, Schmid DM, Gaul G, Hauriet D. Botulinum-A toxin for treating detrusor hyperre-flexia in spinal cord injured patients: a new alternative to anticholinergic drugs? Preliminary results. J Urol 2000;164:692-7.
. Klaphajone J, Kitisomprayoonkul W, Sriplakit S. Botulinum toxin type A injections for treating neurogenic detrusor overactivity combined with low-compliance bladder in patients with spinal cord lesions. Arch Phys Med Rehabil 2000; 86: 2114–8.
. Goldwasser B, Webster GD. Augmentation and substitution enterocystoplasty. J Urol 1986﹔135:215-224.
Point 7: Please provide references for lines 257-259 and 261-263.
Response 7: Thank you for your comment. We have provided the references “Weld KJ, Graney MJ, Dmochowski RR. Differences in bladder compliance with time and associations of bladder management with compliance in spinal cord injured patients. J Urol. 2000 Apr;163(4):1228-33.” for lines 257-259 (line 307-308 in the revised mauscript).
We have provided the references “
. Schulte-Baukloh H, Weiss C, Stolze T, Herholz J, Sturzebecher B, Miller K, et al: Botulinum-A toxin detrusor and sphincter injection in treatment of overactive bladder syndrome: objective outcome and patient satisfaction. Eur Urol 2005;48:984-90.
. Klaphajone J, Kitisomprayoonkul W, Sriplakit S. Botulinum toxin type A injections for treating neurogenic detrusor overactivity combined with low-compliance bladder in patients with spinal cord lesions. Arch Phys Med Rehabil 2000; 86: 2114–8. “ for line 261-263 (line 310-312 in the revised manuscript).
Point 8: Please provide references for lines 309-316.
Response 8: Thank you for your comment. We have provided the references as below in the revised manuscript.
. Schurch B, Stöhrer M, Kramer G, Schmid DM, Gaul G, Hauri D. Botulinum-A toxin for treating detrusor hyperreflexia in spinal cord injured patients: A new alternative to anticholinergic drugs? Preliminary results. J Urol. 2000;164:692–7.
. Kuo HC. Urodynamic evidence of effectiveness of botulinum A toxin injection in treatment of detrusor overactivity refractory to anticholinergic agents. Urology. 2004;63:868–72.
. Goldwasser B, Webster GD. Augmentation and substitution enterocystoplasty. J Urol 1986﹔135:215-224.
. Blok BF, Castro‐Diaz D, Del Popolo G, et al. EAU Guidelines on Neuro‐Urology. European Association of Urology. 2019.
Reviewer 2 Report
Main comments:
1. I found the manuscript title to be suboptimal. If I understand the manuscript correctly, it is not about bladder management AND urological complications but about management strategies to avoid such complications. Perhaps a reworded title is more appropriate.
2. Section 3: At least in my country, probiotics are gaining relevance in the prevention and management of UTI as they can be similarly effective but cause much less resistance development. Would you care to comment?
3. There is some recent work on beta-3 adrenoceptor agonists (mirabegron, vibegron) in NDO. Is that worth mentioning?
4. Overall, I tend to concur with the statements and recommendations in the manuscript. However, there are two areas in which I would like to see more information and discussion: A) Multiple options of management are mentioned for several complications. However, it does not become clear whether some are preferred and others are backup, or all a similarly recommended. B) The authors mention at some points that the SCI group at large is heterogeneous. However, they say little about whether this affects management recommendations. If it does not in their view, that could also be stated more explicitly.
Other comments:
5. L. 48: I suggest replacing “significant”. Apparently, the authors think about the plain English meaning such as “relevant”, but a reader may misinterpret this as talking about a p-value. See also l. 81 and other places throughout the manuscript.
6. L. 49-50: Does the failure to store or empty lead to poor bladder compliance or shouldn’t this be the other way round? The other factors of course are correctly mentioned as results of storage and voiding impairment.
7. The urological complications listed in l. 50-54 differ from those listed in l. 81-83. Shouldn’t this be consistent? Perhaps even more importantly, renal failure (which I consider very important) is only mentioned in l. 81-83.
8. L. 83: The abbreviation UI has not been introduced. Either do so here or replace by UUI that has been introduced, if that is what you wish to talk about.
9. L. 109-110: Not sure whether this statement is helpful. After all, people without SCI also have an almost 100% chance to experience a UTI within a 40 year period (at least the women). I suggest rewording to point better to the specific risk in SCI.
10. L. 161: Isn’t the 1st statement here trivial after all we have read in the previous sections?
11. L. 265-266: How does this compare to the incidence in non-SCI people? A 3% chance within 10 years may not differ that much from the general population. Moreover, this text is redundant with that in l. 280-281; one of them can be deleted.
Author Response
Dear reviewer:
I am very grateful to your comments for the manuscript. According with your advice, we amended the relevant part in manuscript. The reviewer’s questions were answered below.
Reviewer 2.
Point 1: I found the manuscript title to be suboptimal. If I understand the manuscript correctly, it is not about bladder management AND urological complications but about management strategies to avoid such complications. Perhaps a reworded title is more appropriate.
Response 1: Thank you for your suggestion. We have changed the title to “Bladder Management Strategies on the Urological complications in Patients with Chronic Spinal Cord Injury”.
Point 2: Section 3: At least in my country, probiotics are gaining relevance in the prevention and management of UTI as they can be similarly effective but cause much less resistance development. Would you care to comment?
Response 2: Thank you for your great comment. We have discussed more about recurrent UTI as below in the revised manuscript.
Recurrent UTI in SCI patients may indicate suboptimal management of NLUTD and its subsequent urological complications. Besides to botulinum toxin A (BoNT-A) for NDO [1], removal of bladder stones and avoiding indwelling catheters if possible [2], there is still limited evidence for preventive strategies currently. Regarding to prophy-lactic antibiotics, daily antibiotic prophylaxis is not suggested for the prevention of re-current UTI [3], while weekly oral cyclic antibiotic (WOCA) showed significantly reduced UTI without emergence of bacterial resistance compared to control group [4, 5]. Probiotics may be a potential strategy to reduce recurrent UTI; however, in the recent systemic review [6], only two studies showed that probiotics could reduce risk of recurrent UTIs and the remainder demonstrated inconclusive results. Further large randomized studies were needed to comment on the effect of the WOCA and probiot-ics.
Reference:
- Jia, C., et al. Detrusor botulinum toxin A injection significantly decreased urinary tract infection in patients with traumatic spinal cord injury. Spinal Cord, 2013. 51: 487.
- Biering-Sorensen, F., et al. Urinary tract infections in patients with spinal cord lesions: treatment and prevention. Drugs, 2001. 61: 1275.
- Garibaldi RA, et al. Factors predisposing to bacteriuria during indwelling urethral catheterization. N Engl J Med 1974; 291:215–9.
- Dinh A, et al. Weekly Sequential Antibioprophylaxis for Recurrent Urinary Tract Infections Among Patients With Neurogenic Bladder: A Randomized Controlled Trial. Clin Infect Dis. 2020 Dec 15;71(12):3128-3135
- Salomon J, et al. Prevention of urinary tract infection in spinal cord-injured patients: safety and efficacy of a weekly oral cyclic antibiotic (WOCA) programme with a 2 year follow-up—an observational prospective study. J Antimicrob Chemother 2006; 57:784–8
- New FJ, et al. Role of Probiotics for Recurrent UTIs in the Twenty-First Century: a Systematic Review of Literature. Curr Urol Rep. 2022 Feb;23(2):19-28.
Point 3: There is some recent work on beta-3 adrenoceptor agonists (mirabegron, vibegron) in NDO. Is that worth mentioning?
Response 3: Thank you for your great comment. We have discussed more as below in the revised manusciprt.
Regarding to the medication for NDO, antimuscarinics are the most common treatment and suggested as the first-line choice by current guidelines [1]. The role of Beta-3-adrenergic receptor agonists, which is not yet approved by the FDA for neurogenic bladder, is still unclear. In a recent systemic review, mirabegron could improve the storage symptoms of NLUTD and urologic QoL with very few side effects [2]. Improved maximum cystometric capacity and bladder compliance were demonstrated after treated with mirabegron in only two studies with short follow-up [3, 4]; however, the other studies showed no significant changes of urodynamic pa-rameters [5, 6]. Vibegron, another novel Beta-3-adrenergic receptor agonists, was reported to improve bladder capacity and bladder compliance without apparent ad-verse effects in SCI patients with NLUTD in two recent studies with limited cases [7, 8]. Further prospective studies are necessary for role of Beta-3-adrenergic receptor agonists in the treatment for NDO.
Reference:
- Blok BF, Castro‐Diaz D, Del Popolo G, et al. EAU Guidelines on Neuro‐Urology. European Association of Urology. 2019.
- El Helou E, Labaki C, Chebel R, El Helou J, Abi Tayeh G, Jalkh G, Nemr E. The use of mirabegron in neurogenic bladder: a systematic review. World J Urol. 2020 Oct;38(10):2435-2442.
- Park JS, Lee YS, Lee CN, Kim SH, Kim SW, Han SW. Efficacy and safety of mirabegron, a β3-adrenoceptor agonist, for treating neurogenic bladder in pediatric patients with spina bifida: a retrospective pilot study. World J Urol. 2019 Aug;37(8):1665-1670.
- Krhut J, Borovička V, Bílková K, Sýkora R, Míka D, Mokriš J, Zachoval R. Efficacy and safety of mirabegron for the treatment of neurogenic detrusor overactivity-Prospective, randomized, double-blind, placebo-controlled study. Neurourol Urodyn. 2018 Sep;37(7):2226-2233.
- Welk B, Hickling D, McKibbon M, Radomski S, Ethans K. A pilot randomized-controlled trial of the urodynamic efficacy of mirabegron for patients with neurogenic lower urinary tract dysfunction. Neurourol Urodyn. 2018 Nov;37(8):2810-2817.
- Wöllner J, Pannek J. Initial experience with the treatment of neurogenic detrusor overactivity with a new β-3 agonist (mirabegron) in patients with spinal cord injury. Spinal Cord. 2016 Jan;54(1):78-82
- Aoki K, Momose H, Gotoh D, Morizawa Y, Hori S, Nakai Y, Miyake M, Anai S, Torimoto K, Tanaka N, Yoneda T, Matsumoto Y, Fujimoto K. Video-urodynamic effects of vibegron, a new selective β3-adrenoceptor agonist, on antimuscarinic-resistant neurogenic bladder dysfunction in patients with spina bifida. Int J Urol. 2022 Jan;29(1):76-81
- Matsuda K, Teruya K, Uemura O. Urodynamic effect of vibegron on neurogenic lower urinary tract dysfunction in individuals with spinal cord injury: A retrospective study. Spinal Cord. 2022 Aug;60(8):716-721.
Point 4: Overall, I tend to concur with the statements and recommendations in the manuscript. However, there are two areas in which I would like to see more information and discussion: A) Multiple options of management are mentioned for several complications. However, it does not become clear whether some are preferred and others are backup, or all a similarly recommended. B) The authors mention at some points that the SCI group at large is heterogeneous. However, they say little about whether this affects management recommendations. If it does not in their view, that could also be stated more explicitly.
Response 4: Thank you for your great comment.
(A) We briefly concluded it in the conclusion section in the revised manuscript as below. The ideal bladder management strategy in patients with chronic SCI has been the sub-ject of much debate among urologists. Generally, CIC is the preferred management for SCI patients who failure to empty their bladders effectively [130] and indwelling tran-surethral and suprapubic catheter are suggested to be avoided whenever possible. However, the bladder treatment strategy should be flexible to every patient. The priorities in the management of NLUTD in SCI patients should be the following in order of importance: (1) preservation of renal function, (2) reduction of UTI, (3) efficient bladder emptying, (4) avoidance of indwelling catheters, (5) patient agreement with management strategy, and (6) freedom from medication after proper bladder management.
(B) In our persent manuscript, we tried to provide the current information about different bladder managements in every urological complication. However, due to different levels and locations of injuries in SCI patients in those retrospective studies, the heterogenenous exist, which were the limitations in most studies. As the result, the ideal bladder management strategy in patients with chronic SCI has been the subject of much debate among urologists. Every SCI patient differs and every SCI patient’s compliance differs. As we mentioned, the bladder treatment strategy should be individualized to every patient, but those strategies share the same prorities and goals.
Point 5: L. 48: I suggest replacing “significant”. Apparently, the authors think about the plain English meaning such as “relevant”, but a reader may misinterpret this as talking about a p-value. See also l. 81 and other places throughout the manuscript.
Response 5: Thank you for your delicated comment. We have addressed our words more carefully in line 50, line 83, and line 350.
Point 6: L. 49-50: Does the failure to store or empty lead to poor bladder compliance or shouldn’t this be the other way round? The other factors of course are correctly mentioned as results of storage and voiding impairment.
Response 6: Thank you for the comment. Poor bladder compliance is one of the consequences in high risk SCI patients with NLUTD if their bladder or bladder outlet dysfunction are not properly managed. We have revised the statement to: “The main problems associated with NLUTD in chronic SCI patients are failure to store or empty or a combination of the two. If the bladder and bladder outlet dysfunction are not properly managed, they may consequently lead to urinary tract infection (UTI), urosepsis, poor bladder compliance, upper urinary tract deterioration, renal failure, urinary tract calculi, autonomic hyperreflexia (dysreflexia), skin complications, depression (which also complicates urologic treatment), and significant morbidity and mortality in some SCI patients [1].”
Reference:
- Ku JH. The management of neurogenic bladder and quality of life in spinal cord injury. BJU Int 2006; 98: 739-745
Point 7: The urological complications listed in l. 50-54 differ from those listed in l. 81-83. Shouldn’t this be consistent? Perhaps even more importantly, renal failure (which I consider very important) is only mentioned in l. 81-83.
Response 7: Thank you for your comment. Indeed, renal failure should be added in line 54. We haved added in the reivsesd manuscript.
Point 8: L. 83: The abbreviation UI has not been introduced. Either do so here or replace by UUI that has been introduced, if that is what you wish to talk about.
Response 8: Thank you for your comment. This should be replaced by UUI. We have revised it in the reivsesd manuscript.
Point 9: L. 109-110: Not sure whether this statement is helpful. After all, people without SCI also have an almost 100% chance to experience a UTI within a 40 year period (at least the women). I suggest rewording to point better to the specific risk in SCI.
Response 9: Thank you for your comment. We have deleted this sentence, and revised the statement to “UTI is the most common reasons for SCI patients presenting to emergency de-partment and re-hospitalized [1].” to emphasize the importance of UTI in SCI patients.
Reference:
- Cardenas DD, Hoffman JM, Kirshblum S, McKinley W. Etiology and incidence of rehospitalization after traumatic spinal cord injury: a multicenter analysis. Arch Phys Med Rehabil. 2004 Nov;85(11):1757-63.
Point 10: L. 161: Isn’t the 1st statement here trivial after all we have read in the previous sections?
Response 10: Thank you for your comment. We have deleted this sentence in the reivsesd manuscript.
Point 11: L. 265-266: How does this compare to the incidence in non-SCI people? A 3% chance within 10 years may not differ that much from the general population. Moreover, this text is redundant with that in l. 280-281; one of them can be deleted.
Response 11: Thank you for your comment. Line 265-266 (line 314-315 in revised manuscript) is the prevalance of kidney stone, while line 280-281 (line 329-330 in revised manuscript) is the incidence of bladder stone. Due to the limitations of the retrospective studies with SCI cohort [1, 2, 3], they didn’t compare the SCI cohort to the general population. Thank you.
Reference:
1, Post M, Noreau L. Quality of life after spinal cord injury. J Neurol Phys Ther. 2005 Sep;29(3):139-46.
- Lane GI, Roberts WW, Mann R, O'Dell D, Stoffel JT, Clemens JQ, Cameron AP. Outcomes of renal calculi in patients with spinal cord injury. Neurourol Urodyn. 2019 Sep;38(7):1901-1906. doi: 10.1002/nau.24091.
- Chen Y, DeVivo MJ, Roseman JM. Current trend and risk factors for kidney stones in persons with spinal cord injury: a longitudinal study. Spinal Cord. 2000;38(6):346‐353
Reviewer 3 Report
This review aims to 'to review the available data to address issues including long-term urological complications in chronic SCI patients, the effects of different bladder management strategies on 76 UTIs, urge urinary incontinence (UUI), poor bladder compliance, preservation of renal 77 function, prevention of urolithiasis, and satisfaction with enterocystoplasty' which is quite extensive.
It is nicely written but my reasons to recommend rejection are:
- It contributes very little to what is already known on the subject and is available in guidelines such as those from the EAU
- The article covers the more mundane aspects of neurourology and does not discuss the more contraversial and interesting areas such as the management of recurrent UTIs in this population, alternatives to anticholinergics such as beta 3 agonists (no mention at all), upfront Botox rather than trials of anticholinergics, early resconstructive surgical management vs repeated Botox for the treatment of poor compliance and associated quality of life implications, neuromodulation, sacral deafferation and sacral anterior root stimulation.
- There are several repetitions throughout the paper
- Some recommendations are also not backed up by evidence and guidelines.
Some specific issues:
Introduction:
- line 48: this assertion needs a reference.
- line: 56: the EAU goals of treatment are different. Protecting the upper tract is the most important goal of the neurourologist
- line 67: the are several instance where crede voiding is advised against
2. Urological complications in chronic SCI patients:
- line 81: this is a repetition
- line 86: I agree with this assertion but it contradicts figure 2.
- Figure 1: this is mostly very helpful but urinary stasis is also likely to contribute to bladder stone formation
3. The effect of different bladder managements on UTI post SCI:
- line 136: why 'in contrast'?. This statement is in agreement
- I disagree with this. ISC is helpful in patients who cannot empty fully and have symptoms or raised filling pressures. It should not just be for those with low leak point pressures.
- there is very little or any advice on the treatment of recurrent UTIs. This is a very important area
4. Bladder management of UUI in chronic SCI patients
- line 161. This is repetition
5. Different types of bladder management of poor compliance:
- poor compliance is a storage issue. CIC aids voiding. It may not aid compliance.
- line 204: mo mention of beta 3 agonists.
- line206: I am not aware of Ca channel blockers as a treatment for overactive bladder.
- 'detrusor hyper-reflexia' is an antiquated term. The correct term is neurogenic detrusor overactivity as per the ICS.
- there is no mention of a catheterisable channels such as Mitrofanoff.
8:
- Repeat Botox is normally required every 6 months for neuropaths and every 9 months for idiopaths.
- IC often required post cystoplasty.
Author Response
Dear reviewer:
I am very grateful to your comments for the manuscript. According with your advice, we amended the relevant part in manuscript. The reviewer’s questions were answered below.
Reviewer 3.
Point 1: Introduction: line 48: this assertion needs a reference.
- line 67: the are several instance where crede voiding is advised against
Response 1: Thank you for your suggestion. We have added the reference in the revised manuscript [1].
Reference 1: Ku JH. The management of neurogenic bladder and quality of life in spinal cord injury. BJU Int 2006; 98: 739-745
Point 2: Introduction: line: 56: the EAU goals of treatment are different. Protecting the upper tract is the most important goal of the neurourologist
Response 2: Thank you for your comment. We are sorry for the order of the goals. Indeed the order of the goals should be protection of the UUT, achievement (or maintenance) of urinary continence, restoration of LUT function and improvement of the patient’s QoL. We have revised in the revised manuscript.
Point 3: Introduction: line 67: the are several instance where crede voiding is advised against
Response 3: Thank you for your comment. Indeed, assisted bladder emptying including Credé manoeuvre, Valsalva manoeuvre and triggered reflex voiding should be discouraged unless the intravesical pressure remains within safe limits [1]. We have discussed more in the revised manuscript.
Reference 1: Apostolidis, A., et al., Neurologic Urinary and Faecal Incontinence, in Incontinence 6th Edition, P. Abrams, L. Cardozo, S. Khoury & A. Wein, Editors. 2017.
Point 4: line 81: this is a repetition
Response 4: Thank you for your comment. We have deleted this sentence in the revised manuscript.
Point 5: line 86: I agree with this assertion but it contradicts figure 2.
Response 5: Thank you for your comment. We have deleted this sentence “Current evidence indicates that the effects on bladder function depend on the different levels and locations of SCI” in the revised manuscript, because this section is discussing about the “Urological complications in chronic SCI patients” and the figure 2 showed the percentage of urological complications based on different levels of spinal cord injuries.
Point 6: Figure 1: this is mostly very helpful but urinary stasis is also likely to contribute to bladder stone formation
Response 6: Thank you for your comment. We have redrawn the figure 1 and added urinary stasis in figure 1.
Point 7: line 136: why 'in contrast'?. This statement is in agreement
Response 7: Thank you for your comment. We have deleted “in contrast to those guideline” in the revised manuscript.
Point 8: I disagree with this. ISC is helpful in patients who cannot empty fully and have symptoms or raised filling pressures. It should not just be for those with low leak point pressures.
Response 8: Thank you for the comment. We agree that CIC is helpful in patients with chronic SCI who cannot empty fully. This section mentioned that SCI patients who undergo bladder management other than spontaneous voiding might have a higher rate of UTI, therefore, if SCI patients could be trained to void spontaneously by straining or percussion, the rate of UTI will decrease. The statement have been revised to: “Overall, we suggest that SCI patients should be treated with CIC rather than a chronic indwelling catheter if they could not empty fully [1]. If possible, the SCI patients with a leak point pressure lower than 40 cmH2O should be trained to void spontaneously by abdominal straining or percussion to lower the rate of UTI or urological complication.”
Reference 1: Weld KJ, Dmochowski RR. Effect of bladder management on urological complications in spinal cord injured patients. J Urol. 2000 Mar;163(3):768-72.
Point 9: There is very little or any advice on the treatment of recurrent UTIs. This is a very important area
Response 9: Thank you for your great comment. We have discussed more as below in the revised manuscript.
Recurrent UTI in SCI patients may indicate suboptimal management of NLUTD and its subsequent urological complications. Besides to botulinum toxin A (BoNT-A) for NDO [1], removal of bladder stones and avoiding indwelling catheters if possible [2], there is still limited evidence for preventive strategies currently. Regarding to prophy-lactic antibiotics, daily antibiotic prophylaxis is not suggested for the prevention of re-current UTI [3], while weekly oral cyclic antibiotic (WOCA) showed significantly reduced UTI without emergence of bacterial resistance compared to control group [4, 5]. Probiotics may be a potential strategy to reduce recurrent UTI; however, in the recent systemic review [6], only two studies showed that probiotics could reduce risk of recurrent UTIs and the remainder demonstrated inconclusive results. Further large randomised studies were needed to comment on the effect of the WOCA and probiot-ics.
Reference:
- Jia, C., et al. Detrusor botulinum toxin A injection significantly decreased urinary tract infection in patients with traumatic spinal cord injury. Spinal Cord, 2013. 51: 487.
- Biering-Sorensen, F., et al. Urinary tract infections in patients with spinal cord lesions: treatment and prevention. Drugs, 2001. 61: 1275.
- Garibaldi RA, et al. Factors predisposing to bacteriuria during indwelling urethral catheterization. N Engl J Med 1974; 291:215–9.
- Dinh A, et al. Weekly Sequential Antibioprophylaxis for Recurrent Urinary Tract Infections Among Patients With Neurogenic Bladder: A Randomized Controlled Trial. Clin Infect Dis. 2020 Dec 15;71(12):3128-3135
- Salomon J, et al. Prevention of urinary tract infection in spinal cord-injured patients: safety and efficacy of a weekly oral cyclic antibiotic (WOCA) programme with a 2 year follow-up—an observational prospective study. J Antimicrob Chemother 2006; 57:784–8
- New FJ, et al. Role of Probiotics for Recurrent UTIs in the Twenty-First Century: a Systematic Review of Literature. Curr Urol Rep. 2022 Feb;23(2):19-28.
Point 10: line 161. This is repetition
Response 10: Thank you for your comment. We have deleted the sentence in the revised manuscript.
Point 11: Poor compliance is a storage issue. CIC aids voiding. It may not aid compliance.
Response 11: Thank you for the comment. This statement is based on the report of reference 1 that poor bladder compliance or its development with time was associated with SCI in patients using chronic transurethral catheter, instead, CIC and spontaneous voiding could sustain normal bladder compliance. We have revised the statement in this section as “Overall, chronic SCI patients with bladder management of CIC or spontaneous voiding could sustain a normal bladder compliance. In contrast, poor bladder compliance or its development with time was associated with SCI patients using chronic transurethral catheters [1].”
Reference 1: Weld KJ, Graney MJ, Dmochowski RR. Differences in bladder compliance with time and associations of bladder man-agement with compliance in spinal cord injured patients. J Urol. 2000 Apr;163(4):1228-33.
Point 12: line 204: no mention of beta 3 agonists.
Response 12: Thank you for your great comment. We have discussed more as below in the revised manusciprt.
Regarding to the medication for NDO, antimuscarinics are the most common treatment and suggested as the first-line choice by current guidelines [1]. The role of Beta-3-adrenergic receptor agonists, which is not yet approved by the FDA for neurogenic bladder, is still unclear. In a recent systemic review, mirabegron could improve the storage symptoms of NLUTD and urologic QoL with very few side effects [2]. Improved maximum cystometric capacity and bladder compliance were demonstrated after treated with mirabegron in only two studies with short follow-up [3, 4]; however, the other studies showed no significant changes of urodynamic pa-rameters [5, 6]. Vibegron, another novel Beta-3-adrenergic receptor agonists, was reported to improve bladder capacity and bladder compliance without apparent ad-verse effects in SCI patients with NLUTD in two recent studies with limited cases [7, 8]. Further prospective studies are necessary for role of Beta-3-adrenergic receptor agonists in the treatment for NDO.
Reference:
- Blok BF, Castro‐Diaz D, Del Popolo G, et al. EAU Guidelines on Neuro‐Urology. European Association of Urology. 2019.
- El Helou E, Labaki C, Chebel R, El Helou J, Abi Tayeh G, Jalkh G, Nemr E. The use of mirabegron in neurogenic bladder: a systematic review. World J Urol. 2020 Oct;38(10):2435-2442.
- Park JS, Lee YS, Lee CN, Kim SH, Kim SW, Han SW. Efficacy and safety of mirabegron, a β3-adrenoceptor agonist, for treating neurogenic bladder in pediatric patients with spina bifida: a retrospective pilot study. World J Urol. 2019 Aug;37(8):1665-1670.
- Krhut J, Borovička V, Bílková K, Sýkora R, Míka D, Mokriš J, Zachoval R. Efficacy and safety of mirabegron for the treatment of neurogenic detrusor overactivity-Prospective, randomized, double-blind, placebo-controlled study. Neurourol Urodyn. 2018 Sep;37(7):2226-2233.
- Welk B, Hickling D, McKibbon M, Radomski S, Ethans K. A pilot randomized-controlled trial of the urodynamic efficacy of mirabegron for patients with neurogenic lower urinary tract dysfunction. Neurourol Urodyn. 2018 Nov;37(8):2810-2817.
- Wöllner J, Pannek J. Initial experience with the treatment of neurogenic detrusor overactivity with a new β-3 agonist (mirabegron) in patients with spinal cord injury. Spinal Cord. 2016 Jan;54(1):78-82
- Aoki K, Momose H, Gotoh D, Morizawa Y, Hori S, Nakai Y, Miyake M, Anai S, Torimoto K, Tanaka N, Yoneda T, Matsumoto Y, Fujimoto K. Video-urodynamic effects of vibegron, a new selective β3-adrenoceptor agonist, on antimuscarinic-resistant neurogenic bladder dysfunction in patients with spina bifida. Int J Urol. 2022 Jan;29(1):76-81
- Matsuda K, Teruya K, Uemura O. Urodynamic effect of vibegron on neurogenic lower urinary tract dysfunction in individuals with spinal cord injury: A retrospective study. Spinal Cord. 2022 Aug;60(8):716-721.
Point 13: line206: I am not aware of Ca channel blockers as a treatment for overactive bladder.
Response 13: We are sorry for the mistake. We have deleted the calcium channel blockers in the revised manuscript.
Point 14: 'detrusor hyper-reflexia' is an antiquated term. The correct term is neurogenic detrusor overactivity as per the ICS.
Response 14: Thank you for your comment. We have revised to NDO in the revised manuscript.
Point 15: there is no mention of a catheterisable channels such as Mitrofanoff.
Response 15: Thank you for your comment. Indeed, when no other therapy is successful, urinary diversion should be considered. The continent diversion includes the creation of pouches or the addition of a catheterizable channel by Mitrofanoff appendicovesicostomy. We have added them in the revised manuscript.
Point 16: Repeat Botox is normally required every 6 months for neuropaths and every 9 months for idiopaths.
Response 16: Thank you for your comment. In our manuscript, we mentioned that “Repeat BoNT-A injections every 6–9 months are necessary to maintain the therapeutic effects on NDO, especially in patients with chronic SCI.”. This sentence is according to the below studies. In 2000, Schurch et al. [1] published the first report on the injection of botulinum toxin type A (Botox, Allergan® 200/300 units) into the detrusor muscle of patients with SCI for the treatment of NDO. The authors reported the induced bladder paresis to persist for at least 9 months. Our previous study found the regained urinary continence or improved urgency of single detrusor injection of 200 U of BoNT-A to be 73.3% among patients with SCI and NDO, with therapeutic effects lasting 3–9 months (mean, 5.3 months) [2]. We have added the reference in the revised manuscript.
Reference:
- Schurch B, Stöhrer M, Kramer G, Schmid DM, Gaul G, Hauri D. Botulinum-A toxin for treating detrusor hyperreflexia in spinal cord injured patients: A new alternative to anticholinergic drugs? Preliminary results. J Urol. 2000;164:692–7.
- Kuo HC. Urodynamic evidence of effectiveness of botulinum A toxin injection in treatment of detrusor overactivity refractory to anticholinergic agents. Urology. 2004;63:868–72.
Point 17: IC often required post cystoplasty.
Response 17: Thank you for your comment. Indeed, we agreed with you, as our statement in the present manuscript, “76% of patients after bladder augmentation required CIC, while others could void spontaneously with the Credé maneuver.“. Bladder augmentation is a highly successful procedure that stabilises renal function and prevents anatomical deterioration, while intermittent catheterisation may still become necessary after this procedure.
Round 2
Reviewer 3 Report
The abstract needs rewriting to reflect EAU treatment aims.
- Introduction: 1. Autonomic dystreflexia is not a consequence of poor bladder management. It is a consequence of spinal cord injury (line 55). 2. Spinal cord injury does not inevitably result in NLUTD as it can be partial. Complete SCI does.
- 3. The effect of different bladder managements on UTI post-SCI: if a review is being attempted here, D-mannose, cranberry supplementation, oestrogen supplementation, Methenamine Hippurate, UTI vaccines, intravesical cystostat/Hyacyst/ ialuril and intravesical gentamicin will all need to be discussed.
- 4. Bladder management of urge urinary incontinence (UUI) in chronic SCI patients: intrasphincteric Botox is a contraversial area and will need more extensive discussion.
- 5. Different types of bladder management on poor bladder compliance: Line 241 to 245: This sentence is difficult to understand. Please also do not use the expression 'detrusor hypereflexia'. This isn outdated as previously explained.
- 8. Long-term complication and satisfaction of augmentation enterocystoplasty (AE) in 356 chronic SCI patients. The aims of this paper include the assessment of satisfaction with AE. The paper dedicates one short paragraph to this very interesting area which is the subject of much debate. Which patients benefit? Who would you persuade to undergo AE? IS there a risk of bladder cancer (yes)? Which patients get bladder cancer? How should they be surveryed?
Author Response
Response to Reviewer Comments
Dear reviewer:
I am very grateful to your comments for the manuscript. According with your advice, we amended the relevant part in manuscript. The reviewer’s questions were answered below.
Point 1: The abstract needs rewriting to reflect EAU treatment aims.
Response 1: Thank you very much. According to 2022 EAU guideline [1], the primary aims for treatment of neuro-urological symptoms, and their priorities, are (1) protection of the upper urinary tract (2) achievement (or maintenance) of urinary continence (3) restoration of LUT function (4) improvement of the patient’s QoL. Further considerations are the patient’s disability, cost-effectiveness, technical complexity and possible complications. We have revised our abtract to “ To address neurogenic lower urinary tract dysfunction after spinal cord injury, proper and adequate bladder management is important in spinal cord injury rehabilitation, with the goal and priorities of the protection of upper urinary tract function, maintaining continence, preserving lower urinary tract function, improvement of SCI patients’ quality of life, achieving compatibil-ity with the patient’s lifestyle and decreasing urological complications. “ based on EAU treatment aim.
Reference 1:
- Blok BF, Castro‐Diaz D, Del Popolo G, et al. EAU Guidelines on Neuro‐ European Association of Urology. 2022.
Point 2:
Introduction: 1. Autonomic dystreflexia is not a consequence of poor bladder management. It is a consequence of spinal cord injury (line 55). 2. Spinal cord injury does not inevitably result in NLUTD as it can be partial. Complete SCI does.
Response 2: Thank you very much. 1. Autonomic dystreflexia has been deleted in the revised mauscript. 2. We agreed with your comment. In the manuscript, we mentionsed that “Altered lower urinary tract (LUT) function, known as neurogenic LUT dysfunction (NLUTD) due to central or peripheral neurogenic lesions, frequently occurs secondary to chronic SCI and has an significant impact on quality of life. ” NLUTD may happen after SCI. We didn’t mention that the SCI patient must suffer from NLUTD.
Point 3:
The effect of different bladder managements on UTI post-SCI: if a review is being attempted here, D-mannose, cranberry supplementation, oestrogen supplementation, Methenamine Hippurate, UTI vaccines, intravesical cystostat/Hyacyst/ ialuril and intravesical gentamicin will all need to be discussed.
Response 3: Thank you for your great comment. We have discussed more in the revised manuscript as below. Antibiotic bladder instillations, such as neomycin-polymyxin or gentamicin, can decrease frequency of symptomatic UTIs in neurogenic bladder patients on CIC, without increasing multidrug resistance in UTI organisms [1, 2]. Intravesical hyaluronic acid instillation is efficient and safe in patients having neurogenic bladder [3,4]. However, further large randomized studies were needed to comment on the effect of the WOCA, probiotics, blad-der instillations of either antibiotics or hyaluronic acid. EAU guideline could not find suf-ficient evidence to recommend other prevention methods of UTI such as cranberry juice, methenamine Hippurate , L-methionine, estrogen supplementation or D-mannose in patients with neuro-urological symptoms [5-7].
Reference:
1: Cox L, He C, Bevins J, Clemens JQ, Stoffel JT, Cameron AP. Gentamicin bladder instillations decrease symptomatic urinary tract infections in neurogenic bladder patients on intermittent catheterization. Can Urol Assoc J. 2017 Sep;11(9):E350-E354.
2: Huen KH, Nik-Ahd F, Chen L, Lerman S, Singer J. Neomycin-polymyxin or gentamicin bladder instillations decrease symptomatic urinary tract infections in neurogenic bladder patients on clean intermittent catheterization. J Pediatr Urol. 2019 Apr;15(2):178.e1-178.e7.
3: Ziadeh T, Chebel R, Labaki C, Saliba G, Helou EE. Bladder instillation for urinary tract infection prevention in neurogenic bladder patients practicing clean intermittent catheterization: A systematic review. Urologia. 2022 May;89(2):261-267.
4: King GK, Goodes LM, Hartshorn C, Thavaseelan J, Jonescu S, Watts A, Rawlins M, Woodland P, Synnott EL, Barrett T, Hayne D, Boan P, Dunlop SA. Intravesical hyaluronic acid with chondroitin sulphate to prevent urinary tract infection after spinal cord injury. J Spinal Cord Med. 2022 Jul 6:1-7.
5: Gallien P, Amarenco G, Benoit N, Bonniaud V, Donzé C, Kerdraon J, de Seze M, Denys P, Renault A, Naudet F, Reymann JM. Cranberry versus placebo in the prevention of urinary infections in multiple sclerosis: a multicenter, randomized, placebo-controlled, double-blind trial. Mult Scler. 2014 Aug;20(9):1252-9
6: Lee BS, Bhuta T, Simpson JM, Craig JC. Methenamine hippurate for preventing urinary tract infections. Cochrane Database Syst Rev. 2012 Oct 17;10(10):CD003265.
7: Günther, M., et al. Harnwegsinfektprophylaxe. Urinansäuerung mittels L-Methionin bei neurogener Blasenfunktionsstörung. Urologe B, 2002. 42: 218.
Point 4:
Bladder management of urge urinary incontinence (UUI) in chronic SCI patients: intrasphincteric Botox is a contraversial area and will need more extensive discussion.
Response 4: Thank you for your comment. We have discussed more in the revised manuscript as below. Although the efficacy of BoNT-A over sphincter was reported to be high with few side effects, this treatment is not licensed and the necessity of reinjection is its main disad-vantage [1-3]. In addition, because of limited evidence with small studies, further ran-domized studies were still needed to comment on the effectiveness of BoNT-A and the optimal dose [4].
Reference:
1.Dykstra, D.D., et al. Treatment of detrusor-sphincter dyssynergia with botulinum A toxin: a doubleblind study. Arch Phys Med Rehabil, 1990. 71: 24.
- Schurch, B., et al. Botulinum-A toxin as a treatment of detrusor-sphincter dyssynergia: a prospective study in 24 spinal cord injury patients. J Urol, 1996. 155: 1023.
- Huang, M., et al. Effects of botulinum toxin A injections in spinal cord injury patients with detrusor overactivity and detrusor sphincter dyssynergia. J Rehabil Med, 2016. 48: 683.
- Utomo E, Groen J, Blok BF. Surgical management of functional bladder outlet obstruction in adults with neurogenic bladder dysfunction. Cochrane Database Syst Rev. 2014 May 24;(5):CD004927.
Point 5:
Different types of bladder management on poor bladder compliance: Line 241 to 245: This sentence is difficult to understand. Please also do not use the expression 'detrusor hypereflexia'. This isn outdated as previously explained.
Response 5: Thank you for your comment. We have revised to neurogenic detrusor overactivity (NDO) in the revised manuscript.
Point 6:
- 8. Long-term complication and satisfaction of augmentation enterocystoplasty (AE) in 356 chronic SCI patients. The aims of this paper include the assessment of satisfaction with AE. The paper dedicates one short paragraph to this very interesting area which is the subject of much debate. Which patients benefit? Who would you persuade to undergo AE? IS there a risk of bladder cancer (yes)? Which patients get bladder cancer? How should they be surveryed?
Point 6: Thank you for your great comment. In our manuscript, we have mentioned as below. Currently, the first-line treatment for NLUTD in SCI is anticholinergic agents, timely voiding schedules, and CIC in relatively good circumstances, followed by detrusor BoNT-A injections when the effects of conservative treatment are inadequate. If the outcome is still refractory to repeated BoNT-A injections, intravesical pressure will remain high, which eventually leads to hydronephrosis, renal failure, and UUI, so more aggressive surgical treatment should be considered to obtain life-long therapeutic effects instead of periodic BoNT-A injections.
In addition, we have discussed more in the revised manuscript as below. AE should be considered in patients with reduced bladder capacity and poor compliance due to refractory NLUTD. In addition, AE is associated with a risk of bladder malignancy. A recent systemic review reported that the estimated incidence to develop a malignant tumor after AE ranged from 0 to 272.3 per 100,000 patients/year [1-9] and that 51.6% of the malignancy was ade-nocarcinoma. Up to 90% of bladder malignancy was diagnosed more than 10 years after AE. Although the exact mechanism of carcinogenesis after AE is still unclear, several fac-tors such as bacteriuria, chronic inflammation, urinary hyperosmolality conditions, are reported to be possibly involved [10-15]. The follow-up time of regular surveillance after AE is controversial; however, an annual cystoscopy starting 10 years after AE was rec-ommended by most studies [1, 4, 16, 17].
Reference:
- Ali-El-Dein B, El-Tabey N, Abdel-Latif M, et al. Late uro-ileal cancer after incorporation of ileum into the urinary tract. J Urol 2002;167:84–7.
- Husmann DA, Rathbun SR. Long-term follow up of enteric bladder augmentations: The risk for malignancy. J Pediatr Urol 2008;4:381–5.
- Higuchi TT, Granberg CF, Fox JA, et al. Augmentation cystoplasty and risk of neoplasia: Fact, fiction and controversy. J Urol 2010;184:2492–6.
- Kalble T, Hofmann I, Riedmiller H, et al. Tumor growth in urinary diversion: A multicenter analysis. Eur Urol 2011;60:1081–6.
- Castellan M, Gosalbez R, Perez-Brayfield M, et al. Tumor in bladder reservoir after gastrocystoplasty. J Uro 2007;178:1771–4.
- Vemulakonda VM, Lendvay TS, Shnorhavorian M, et al. Metastatic adenocarcinoma after augmentation gastrocystoplasty. J Urol 2008;179:1094–7.
- Soergel TM, Cain MP, Misseri R, et al. Transitional cell carcinoma of the bladder following augmentation cystoplasty for the neuropathic bladder. J Urol 2004;172:1649–52.
- Kispal Z, Balogh D, Erdei O, et al. Complications after bladder augmentation or substitution in children: A prospective study of 86 patients. BJU International 2011;108:282–9.
- Biardeau X, Chartier-Kastler E, Rouprêt M, Phé V. Risk of malignancy after augmentation cystoplasty: A systematic review. Neurourol Urodyn. 2016 Aug;35(6):675-82.
- Higgy NA, Verma AK, Erturk € E, et al. Escherichia coli infection of the urinary bladder: Induction of tumours in rats receiving nitrosamine precursors and augmentation of bladder carcinogenesis by N-nitrosobutyl (4-hydroxybutyl) amine. IARC Sci Publ 1987;380–3.
- Nurse DE, Mundy AR. Cystoplasty infection and cancer. Neurourol Urodyn 1989;8:343–4.
- Vajda P, Kaiser L, Magyarlaki T, et al. Histological findings after colocystoplasty and gastrocystoplasty. J Urol 2002;168:698–701.
- Austen M, Kalble T. Secondary malignancies in different forms of urinary diversion using isolated gut. J Urol 2004;172:831–8.
- Malone MJ, Izes JK, Hurley LJ. Carcinogenesis: The fate of intestinal segments used in urinary reconstruction. Urol Clin North Am 1997;24:723–28.
- Dixon BP, Chu A, Henry J, et al. Increased cancer risk of augmentation cystoplasty: Possible role for hyperosmolal microenvironment on DNA damage recognition. Mutat Res 2009;670:88–95.
- Hamid R, Greenwell TJ, Nethercliffe JM, et al. Routine surveillance cystoscopy for patients with augmentation and substitution cystoplasty for benign urological conditions: Is it necessary?. BJU Int 2009;104:392–5.
- Shaw J, Lewis MA. Bladder augmentation surgery—What about the malignant risk? Eur J Pediatr Surg 1999;9:39–40.